# Critical role of water structure around interlayer ions for ion storage in layered double hydroxides

Tomohito Sudare [1] ✉, Takuro Yamaguchi[2], Mizuki Ueda[2], Hiromasa Shiiba[1], Hideki Tanaka [1], Mongkol Tipplook[1], Fumitaka Hayashi[2] & Katsuya Teshima [1,2] ✉

Water-containing layered materials have found various applications such as water purification and energy storage. The highly structured water molecules around ions under the confinement between the layers determine the ion storage ability. Yet, the relationship between the configuration of interlayer ions and water structure in high ion storage layered materials is elusive. Herein, using layered double hydroxides, we demonstrate that the water structure is sensitive to the filling density of ions in the interlayer space and governs the ion storage. For ion storage of dilute nitrate ions, a 24% decrease in the filling density increases the nitrate storage capacity by 300%. Quartz crystal microbalance with dissipation monitoring studies, combined with multimodal ex situ experiments and theoretical calculations, reveal that the decreasing filling density effectively facilitates the 2D hydrogen-bond networking structure in water around interlayer nitrate ions along with minimal change in the layered structure, leading to the high storage capacity.

Water molecules structured around ions are ubiquitous not only in bulk solution[1] and at solid/liquid interfaces[2] but also in nanostructures[3–5]. Ions and structured water confined in such nanostructures have found various ion storage applications such as water purification and energy storage[6–9]. Recently, structured water has been claimed to play an important role in endowing layered materials with high ion storage capacities that are governed by the flexibility of the host layers[9,10]. In the interlayer space, water molecules are incorporated into the pores that are not filled with ions, thereby stabilising the layered structure. Therefore, the water structure and content are sensitive to the configuration of the interlayer ions. Although the ion configuration in various crystal structures and reaction intermediates controls the ion storage ability[11–14], it has rarely been systematically investigated.

To study the interplay between structured water and ion configuration, the filling density of ions in the interlayer space is considered. A careful observation of the interlayer structure of layered metal oxides/hydroxides suggests an inverse correlation between the stability of the water and layered structure (Fig. 1)[15]. Considering that the large unfilled pores are enough to form a hydrogen-bond network in interlayer water, water is assumed to structure around ions in the interlayer space[16,17]. In contrast, progressive water structuration involves the expansion of the interlayer space, thereby possibly destabilising the structure. This is expected to be profound in high-filling regimes because of the small original pore. Therefore, an understanding of the water structure at different interlayer filling densities provides an essential basis for the rational design of ion storage materials.

The high tunability of the host charge density and the rigid layer-stacking structure in layered double hydroxides (LDHs) can be effectively utilised for this purpose. LDHs are a conventional anionic clay with the general formula $[M^{2+}_{1-x}M^{3+}_x(OH)_2]^{x+}(A^{n-})_{x/n}\cdot mH_2O$, where $M^{2+}$ and $M^{3+}$ are divalent and trivalent metal cations, respectively, $A^{n-}$ is the charge-compensating anion, and $x$ is the molar fraction of the trivalent cation $[M^{3+}/(M^{2+} + M^{3+}) = 0.166-0.33]$[18–21]. The number and distribution of the positive charges associated with the host layers are controlled by $x$. Therefore, the interlayer filling density of the charge-

[1]Research Initiative for Supra-Materials (RISM), Shinshu University, 4-17-1 Wakasato, Nagano 380-8553, Japan. [2]Department of Materials Chemistry, Faculty of Engineering, Shinshu University, 4-17-1 Wakasato, Nagano 380-8553, Japan. ✉e-mail: tsudare@shinshu-u.ac.jp; teshima@shinshu-u.ac.jp

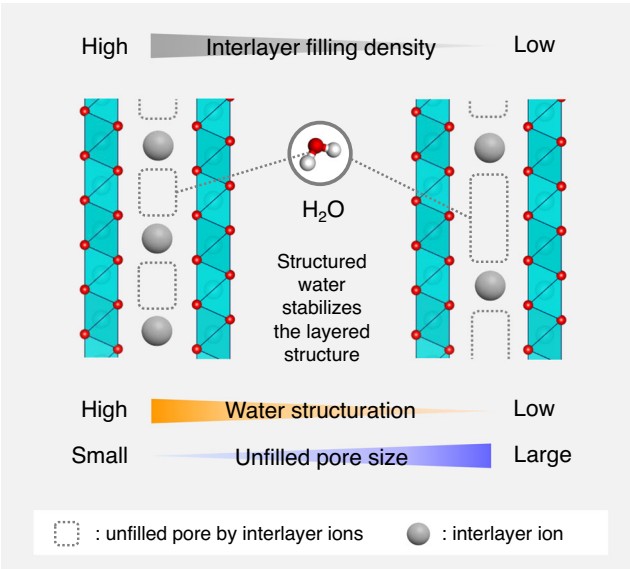

**Fig. 1 | Schematic of the interlayer structure in layered metal oxide/hydroxides with different interlayer filling densities of ions.** The high interlayer filling density of ions results in small pores that are not filled by the interlayer ions; water is incorporated and structured in these pores. The low interlayer filling regime results in large unfilled pores. In all crystal structures, the green octahedra is $MO_6$ or $M(OH)_6$, and the red and white spheres are oxygen or hydroxide ions and hydrogen atoms, respectively. The VESTA programme was used for visualising the crystal structures[15].

compensating counter-anions is precisely determined by $x$. On this basis, we considered the separation of nitrate ($NO_3^-$) ions through the ion-exchange reaction with chloride ions in LDHs. The poor ion storage ability (compared to other anions) for nitrate ion in dilute conditions is the major unsolved issue in the study of LDHs because of the low charge density and nucleophilicity[22], which necessitates a precise control of the binding force between interlayer nitrate ions, water molecules, and the host layers. More importantly, although nitrate and chloride ions have a monovalent negative charge and low nucleophilicity, they possess different molecular structures, resulting in similar interactions between the ions and adsorption site but different hydration behaviours; chloride ions have a high hydration enthalpy[23] and the ability to structure water, as suggested by the Hofmeister series[24]. Thus, the ion-exchange reaction system between nitrate and chloride is a suitable model system for studying the concerted structuring behaviour of water and ions in the two-dimensional confinement of the layered materials. In this study, using two LDHs with different host charge densities ($x$), we firstly demonstrate that the nitrate ion storage capacity is significantly enhanced in a low interlayer filling density. Next, quartz crystal microbalance with dissipation monitoring (QCM-D) measurements were performed to prove the dynamic structural change of the LDHs upon nitrate ion storage. Finally, these results were combined with the multimodal ex situ experimental results and theoretical calculations to clarify the critical role of the water structure on the high ion storage capacity.

## Results and discussion

We prepared two Mg/Al LDHs with different $x$ values by a co-precipitation method followed by chloride ion-exchange treatment to form $Cl^-$-form LDHs[25–27]. The chemical compositions of the LDHs are $[Mg_{0.72}Al_{0.28}(OH)_2](Cl)_{0.28}\cdot0.41H_2O$ and $[Mg_{0.78}Al_{0.22}(OH)_2](Cl)_{0.22}\cdot0.41H_2O$ (determined using the ICP-OES, FT-IR, and TG-DTA techniques; Supplementary Figs. 1, 6a, and 7a), indicating that the interlayer filling densities of ions in the weakly charged latter LDHs are 24% lower than those of the highly charged former LDHs. XRD patterns

(Fig. 2a) and field emission scanning electron microscopy (FE-SEM) images (Fig. 2b) of the as-prepared LDHs indicate that both samples have similar crystallinity and particle size, although the interlayer distance ($d_{003}$) of the highly charged LDHs is smaller by ~0.1 Å than those of the weakly charged LDHs.

Using these two LDHs, the effect of host charge density on the nitrate ion storage capacity was examined. To evaluate the maximum storage capacity ($q_m$), adsorption isotherms were collected at room temperature and fit to appropriate models (Fig. 2c). Two different models were required for valid fitting in the entire range of nitrate concentrations, indicating that nitrate ion-exchange adsorption is driven by different mechanisms based on the extent of reaction. For both LDHs, Langmuir-Freundlich and Langmuir models[28,29] were employed in the dilute (~5.0 mM) and concentrated (~200 mM) regions, respectively (see Supplementary Fig. 2 and Supplementary Table 1). The basic assumptions underlying the Langmuir-Freundlich (energetically heterogeneous adsorption sites) and Langmuir (energetically homogeneous adsorption sites) models imply that nitrate intercalation is accompanied by several metastable interlayer structures and interactions in the early storage regime, and by slight structural changes in the later storage regime. According to the fitting results, the equilibrium constants for the weakly charged LDHs are consistently higher by one order of magnitude than those for the highly charged LDHs in both concentration regions. Moreover, the weakly charged LDHs yield approximately 300% higher nitrate storage capacity in the dilute region compared to the highly charged LDHs, although the nitrate storage capacity decreases in the concentrated region (Fig. 2d). The remarkable difference in the nitrate ion storage ability is attributed to the different intermediate interlayer structures and interactions, which depend on the interlayer filling density.

To further elucidate the mechanism underlying the high nitrate ion storage capacity, we conducted a comparative QCM-D study for the two LDHs (Fig. 3a). QCM-D is powerful in situ measurement tool for understanding ion dynamics of matter. Based on the change in frequency, mass transfer upon intercalation/deintercalation of ion and water has been studied[30–32]. Importantly, dissipation monitoring has been long utilised for studying the dynamic deformation behaviours accompanying viscoelastic changes in water-containing soft matter[33,34]. The viscoelasticity of LDHs represents the overall interlayer interactions. Therefore, by simultaneously monitoring dissipation values and frequencies, the dynamic change in the interlayer interactions along with mass transfer of LDHs upon nitrate intercalation is studied (Fig. 3b). The experiments were carried out with a flow system. The prepared Mg/Al LDHs were deposited on a $SiO_2$-coated Au electrode (Supplementary Fig. 3) and examined by QCM-D. The medium was changed from an aqueous NaCl solution to an aqueous $NaNO_3$ solution, and the changes in frequency and dissipation upon ion exchange were monitored. Prior to the analysis of LDHs, we confirmed that the surface reaction on the $SiO_2$-coated Au electrode has a negligible effect on the QCM-D profiles for the results of the experiments described below (Supplementary Fig. 4).

A distinct difference in the dissipation and frequency changes of the two LDHs is observed in the QCM-D profiles immediately after changing the solution (Fig. 3c; expanded profiles given in Supplementary Fig. 5). Highly charged LDHs show a large decrease in the dissipation and a slight decrease in the frequency, followed by a sudden and large increase in the dissipation and the corresponding large decrease in the frequency. The early change in the dissipation suggests enhanced interlayer interactions triggered by the incorporation of a tiny amount of nitrate in the interlayer space. The slight decrease in the frequency in the early regime may be contributed by the interlayer water (de)intercalation in addition to the mass increase via one-to-one ion exchange between chloride and nitrate. In the later regime, the change in the dissipation and frequency indicates the progressive intercalation of nitrate accompanied by a considerable weakening of

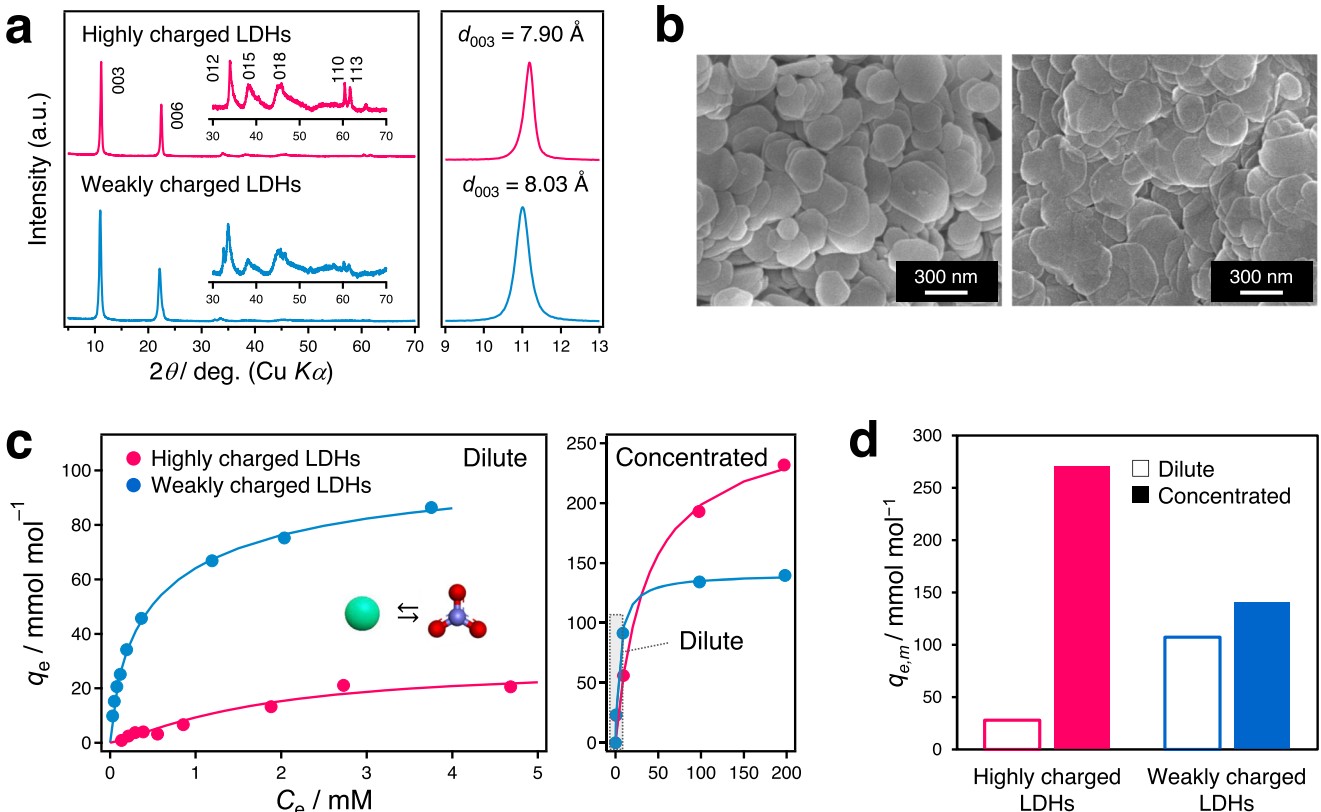

**Fig. 2 | Synthesis of LDHs with different charge densities and their ion storage performance for nitrate. a** Powder X-ray diffraction (XRD) patterns of the as-prepared Mg/Al LDHs with $x = 0.28$ and 0.22. **b** Field emission scanning electron microscopy (FE-SEM) of the as-prepared Mg/Al LDHs with $x = 0.28$ (left) and 0.22 (right). **c** Ion-exchange adsorption isotherms for nitrate with curve-fit using the Langmuir-Freundlich (LF) model in the dilute region (left) and the Langmuir model in the concentrated region (right). **d** Corresponding nitrate ion storage capacity for the as-prepared Mg/Al LDHs with $x = 0.28$ and 0.22. Source data for **a**, **c**, and **d** are provided as a Source Data file.

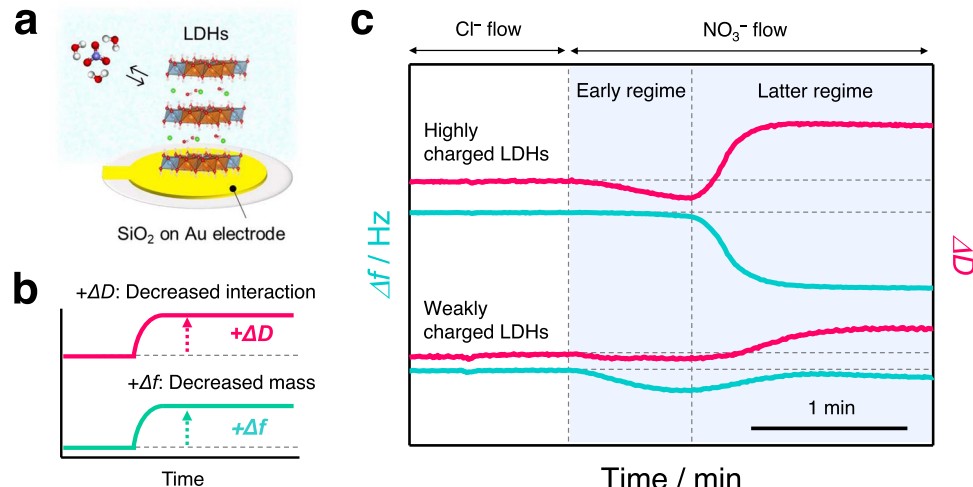

**Fig. 3 | In situ monitoring of interlayer interactions during the ion storage of nitrate in LDHs.** Schematic images of the **a** quartz crystal microbalance with dissipation monitoring (QCM-D) profiles for LDHs and **b** the relationship between the change in the frequency ($\Delta f$) and dissipation ($\Delta D$) and the change in mass and interlayer interaction. **c** The QCM-D profiles for the as-prepared Mg/Al LDH thin films with $x = 0.28$ and 0.22 show the change in the frequency ($\Delta f$) and dissipation ($\Delta D$) upon changing the test solution from aqueous NaCl solution to NaNO₃ solution. The flow rate is set to 300 μL min⁻¹. Source data for **c** are provided as a Source Data file.

the interlayer interactions. The progressive nitrate intercalation across the two regimes drives the transition from strong interlayer interactions to weak interactions in the LDHs, which is energetically unfavourable and results in low nitrate ion storage capacity. This transition accounts for the experimentally observed isotherms. The additional QCM-D measurements at a decreased flow rate, wherein the above-mentioned sudden and large change in the later regime disappeared, support this mechanism (Supplementary Fig. 6). This indicates that the

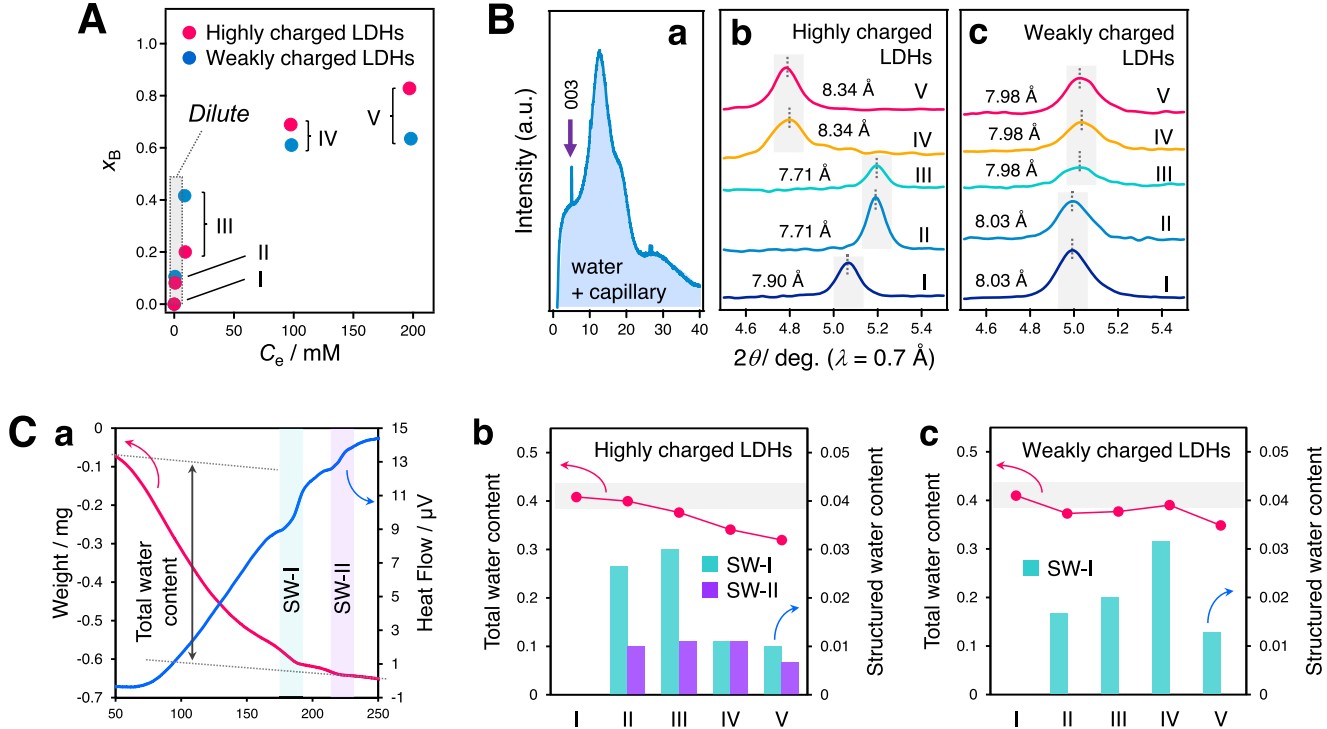

**Fig. 4 | Ex situ analysis of the changes in the interlayer distance and the content of interlayer water in nitrate ion storage. A** Ion-exchange adsorption isotherms for nitrate in the concentrated region with respect to the highly and weakly charged LDHs, corresponding to Fig. 2c after the $Y$ axis is modified to the corresponding $x_B$ value. **B** The representative ex situ synchrotron powder XRD patterns of the highly charged LDHs (II in **A**) (**a**) and closer inspection of the wet-XRD patterns of the

highly charged LDHs (**b**) and weakly charged LDHs (**c**). The Roman numerals I–V correspond to the description of **A. C** The representative thermogravimetry-differential thermal analysis (TG-DTA) profiles of the highly charged LDHs (II in **A**) (**a**) and the total water content and the structured water content in the highly charged LDHs (**b**) and the weakly charged LDHs (**c**). Structured water is referred to as SW. Source data are provided as a Source Data file.

interlayer structures containing excess nitrate observed in the later regime are presumably metastable.

For weakly charged LDHs, a slight decrease in the dissipation is observed (corresponding to slightly enhanced interlayer interactions) along with a large decrease in the frequency (corresponding to the inclusion of a large amount of nitrate into the LDHs) in the early storage regime. In the later regime, a small increase in the dissipation (corresponding to a weakening of the interlayer interactions) and in the frequency value (corresponding to decreasing mass) are observed. This indicates that further intercalation in the latter regime is suppressed due to the weakening of the interlayer interactions and is accompanied by slight dehydration. These comparative QCM-D studies, in combination with the isotherm analysis, reveal that a lower interlayer filling density causes minimal changes in the interlayer interactions, resulting in high nitrate storage capacities under dilute conditions.

To validate the proposed reaction mechanism, structural changes under the ion storage mechanism were analysed. The equilibrated LDHs used in the isotherm measurements (Fig. 2c) were analysed by ex situ synchrotron powder XRD (SPXRD). The re-posted isotherms are given with the Y-axis replaced by the $x_B$ value [= $NO_3^-/(Cl^- + NO_3^-)$] to facilitate discussion (Fig. 4A). Ex situ SPXRD measurements were conducted using slurry-form samples packed in glass capillary, consisting of the LDH powder and the aqueous solution. The obtained pattern consists of the reflections attributed to the LDHs, amorphous water, and the capillary (representative result shown in Fig. 4B-a). A closer inspection of the obtained ex situ SPXRD results for the 003 reflections revealed that highly charged LDHs show contraction and expansion of the interlayer space depending on the $x_B$ value, implying the transition of the orientation of the interlayer nitrate from parallel to tilted toward the *ab*-plane of the host layer. The weakly charged

LDHs show a gradual decrease in the interlayer distance. These results reveal that the contracted (expanded) interlayer space is attributed to the enhanced (weakened) interlayer interactions identified in the QCM-D analysis.

To identify the nature of the interlayer water binding affecting the interlayer interactions, we carried out TG-DTA analysis of the equilibrated LDHs used in the ex situ SPXRD measurements. The typical TG-DTA profiles of the LDHs are shown in Fig. 4C-a (see Supplementary Figs. 7 and 8 for data for other samples). A large loss and two small losses are observed in the TG profile in the temperature range of 50–180 °C and at 180 and 220 °C, respectively, along with the corresponding endothermic peaks in the DTA profile. The former large weight loss is attributed to the content of water physically adsorbed on the particle surface and between particles, and in the interlayer space[35–37]. The latter two peaks are attributed to the structured water strongly bound in the interlayer space with different binding energies. The former and latter structured water are referred to as SW-I and SW-II, respectively. For both LDHs, the total water content of these three features decreased with nitrate intercalation. The structured water emerged only after nitrate intercalation (Figs. 4C-b and 4C-c), indicating that it is established around interlayer nitrates. A remarkable difference is found for the content of structured water. For highly charged LDHs, a large amount of SW-I appeared in the early regime and then the SW-1 amount decreased in the later regime, while for weakly charged LDHs, the amount of SW-1 gradually increased with nitrate intercalation. SW-II appeared only in highly charged LDHs. These results account for the interlayer interactions and structures elucidated by QCM-D and ex situ SPXRD experiments. For highly charged LDHs, the enhanced interlayer interactions in the contracted interlayer space in the early storage regime arise from the presence of strongly bound structured water

(SW-I and -II). The weakened interactions in the expanded interlayer space in the latter regime arise from the disappearance of SW-I. For weakly charged LDHs, a gradual increase in SW-I content facilitates moderate interlayer interactions in the nitrate-incorporated interlayer space, along with minimal structural changes.

Finally, to better understand the role of interlayer water, the variation in energy and the possible interlayer water structures were investigated by DFT calculations. First, we prepared two model LDH structures with highly charged $[Mg_{0.66}Al_{0.33}(OH)_2]$ layers and weakly charged $[Mg_{0.833}Al_{0.166}(OH)_2]$ layers to analyse the above-described experimental results. Second, the interlayer binding energies, the hydrogen-bonding structure associated with the orientation of interlayer anions and water molecules, and the interlayer distance[38] for both LDHs with various interlayer anion compositions ($x_B$) were calculated as a function of the interlayer water content ($m$). The structure with $x_B = 0.125$ in highly charged LDHs and the structure with $x_B = 0.25$ in weakly charged LDHs were employed to investigate the intermediate interlayer structures during the nitrate ion storage. Using the total energy of the optimised structures ($E_{total}$), the binding energy was estimated per interlayer anion ($E_{bind.}$) as:

$$E_{bind.}\left(eV\ anion^{-1}\right) = \{E_{total} : [Mg_{1-x}Al_x(OH)_2](Cl^-)_{1-x_B}(NO_3^-)_{x_B}\bullet mH_2O$$
$$- E_{total} : [Mg_{1-x}Al_x(OH)_2]$$
$$- (1-x_B)E_{total} : Cl^- - x_B\bullet E_{total} : NO_3^-$$
$$- m\bullet E_{total} : H_2O\}/x_B \qquad (1)$$

Finally, to investigate the favourable direction of the ion-exchange system described by $Cl^-(s) + NO_3^-(l) \rightleftharpoons Cl^-(l) + NO_3^-(s)$, we calculated

the binding energy of such system ($E_{sys.}$) using the total energy of the anions and the experimentally obtained hydration enthalpy[23] as defined by,

$$E_{sys.}\left(eV\ anion^{-1}\right)\begin{cases} = E_{bind.} + E_{total} : NO_3^- + \Delta H_{hyd.} : NO_3^- \ (x_B = 0) \\ = E_{bind.} + E_{total} : Cl^- + \Delta H_{hyd.} : Cl^- \ (x_B \neq 0) \end{cases}$$
$$(2)$$

The computational details are described in detail in the Methods section together with supplementary data (Supplementary Figs. 9 and 10). The calculated binding energy of the optimised LDHs with $x_B$ value and the corresponding interlayer distance are given as a function of $m$ in Supplementary Fig. 11. The results revealed that for both LDHs, the interlayer binding energy gradually increased with $m$ and decreased with $x_B$. The interlayer distance shows a gradual increase with $m$ and $x_B$. Interestingly, a drastic increase in the interlayer space is observed for $m = 0.3-0.5$ in highly charged LDHs with $x_B = 0$ and 0.125 (Supplementary Fig. 11b), corresponding to the offset in the binding energy (Supplementary Fig. 11a). A similar behaviour is observed in the weakly charged LDHs with $x_B = 1.0$ (Supplementary Figs. 11c and 11d). These results indicate that such interlayer expansion involves destabilisation of the layered structure.

To describe the stability of the prospective structures more clearly, $E_{sys.}$ values are presented as a function of the interlayer distance ($d$) for highly charged LDHs (Fig. 5a) and for weakly charged LDHs (Fig. 5b). Compared to the weakly charged LDHs, the highly charged LDHs show small negative slopes for all $x_B$ values. This is because the unfilled pores occupied by ions in the interlayer spaces are

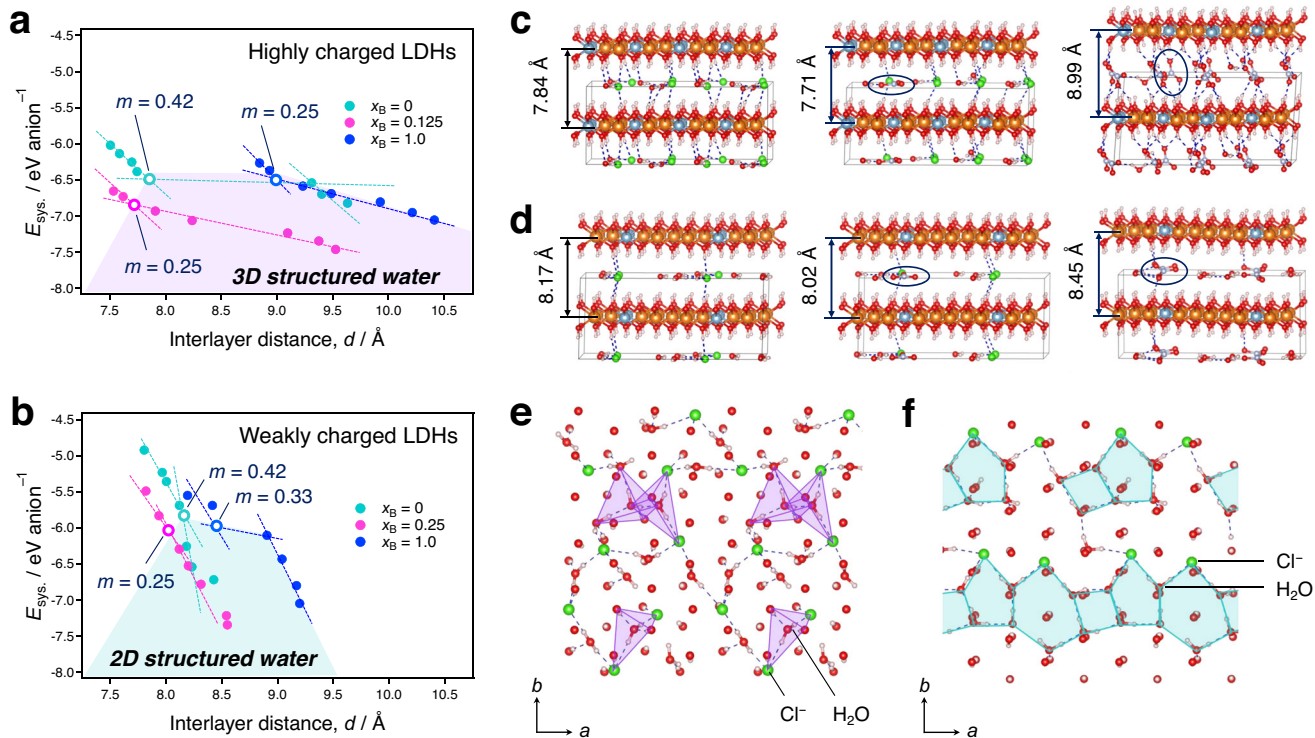

**Fig. 5 | The energy diagram of the nitrate ion storage and the hydrogen-bond structure in the interlayer space.** The calculated binding energy of the ion-exchange system as a function of the interlayer distance of the optimised LDH structure **a** with highly charged LDHs $[Mg_{0.66}Al_{0.33}(OH)_2](NO_3^-)_{xB}$ $(Cl^-)_{1-xB}\bullet mH_2O$ and **b** with weakly charged LDHs: $[Mg_{0.833}Al_{0.166}(OH)_2]$ $(NO_3^-)_{xB}(Cl^-)_{1-xB}\bullet mH_2O$. The optimised LDH structures **c** of highly charged LDHs with $x_B = 0.0$ ($m = 0.42$), $x_B = 0.125$ ($m = 0.25$), and $x_B = 1.0$ ($m = 0.25$), and **d** of weakly charged LDHs with $x_B = 0.0$ ($m = 0.42$), $x_B = 0.25$ ($m = 0.25$), and $x_B = 1.0$ ($m = 0.33$). The three-dimensional and two-dimensional structured

water along with chlorides emerges in the interlayer space of the optimised LDH structure **e** with highly charged LDHs and **f** with weakly charged LDHs, respectively. Here, the molecular arrangements are projected on the $ab$-plane of LDHs. In all crystal structures, orange, pale blue, red, white, green, and blue spheres represent magnesium, aluminium, oxygen, hydrogen, chlorine, and nitrogen atoms, respectively. To describe the interlayer hydrogen-bonding network, the interatomic bond lengths in the interlayer space (dotted lines) are set to 1.88 and 2.30 Å as the maximum H–O and H–Cl bond lengths, respectively[50,51].

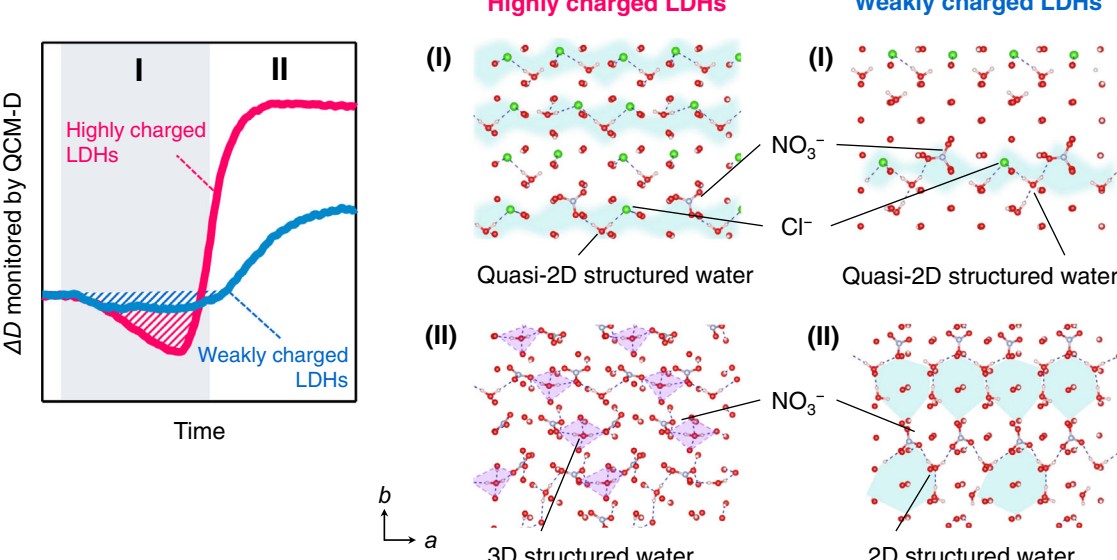

**Fig. 6 | The critical role of interlayer filling density of ions and structured water on nitrate ion storage in LDHs.** The role is further decomposed, combining the obtained QCM-D profiles with the theoretical calculations. In the high interlayer filling density environment of the highly charged LDHs, quasi-2D structured water formed in the early storage regime (I) facilitates strong interlayer interactions, while the 3D structured water in the latter storage regime weakens the interlayer interactions (II). In the low interlayer filling density environment of the weakly charged LDHs, slightly quasi-2D structured water formed in the early storage regime (I) and 2D structured water formed in the latter storage regime (II) facilitate a moderate change in interlayer interactions and progressive nitrate ion intercalation.

of different sizes (the Connolly surface of the dehydrated LDHs is shown in Supplementary Fig. 12). For highly charged LDHs, the energy gain upon filling of the unfilled pores by water molecules is lower than that for the weakly charged LDHs. Remarkably, several points at which two lines intersect are observed. For $x_B = 0$ in highly and weakly charged LDHs, the interlayer distances reproduce the experiments (Supplementary Fig. 10). This implies that the crystal structures obtained at the other specific points can be employed to account for the experimental results (the corresponding interlayer structures are presented in Figs. 5c and 5d). Based on this, the interlayer contraction behaviour in the early nitrate storage regime confirmed experimentally (for both highly and weakly charged LDHs) was reproduced in these calculations. The remarkable contraction-expansion behaviour, along with the transformation of the orientation of the interlayer nitrate from parallel to tilted, was also reproduced in the highly charged LDHs. In this regard, the energetic hierarchy (Fig. 5a) reveals that the energy decreased between $x_B = 0.0$ and $0.125$ and increased between $x_B = 0.125$ and $1.0$ in the highly charged LDHs. On the other hand, in the weakly charged LDHs, the energy decreased between $x_B = 0.0$ and $0.25$ while it remained nearly the same between $x_B = 0.25$ and $1.0$.

Careful inspection of the hydrogen-bonded network structure of water reveals that the water structure determines the specific points (that is, the water content) and hence, the structural stability of the possible structures. For highly charged LDHs, the progressive formation of three-dimensional structured water is confirmed in the high-$m$ region over the above specific points, defining interlayer water connected tetrahedrally to interlayer water molecules, anions, and hydroxide ions in the host layers via hydrogen bonding; its representative structure with $x_B = 0.0$ ($m = 0.58$) is illustrated in Fig. 5e; other structures are shown in Supplementary Figs. 13–18. For weakly charged LDHs, the development of two-dimensional structured water is confirmed in the high-$m$ region; its representative structure with $x_B = 0.0$ ($m = 0.58$) is illustrated in Fig. 5f, confirming the presence of hexagonal, pentagonal, and quadrangular hydrogen-bonding networks. Notably, quadrangular water clusters known as square ice are

rarely found in other materials[3]. The different structural features of the hydrogen-bonded network are attributed to the differently sized pores that are not filled by the interlayer ions – a high (low) filling density tends to facilitate the small (large) pore, which, in turn, leads to the formation of a three (two)-dimensional network along with interlayer ions. Importantly, the advanced water structuring in the high-$m$ region decreases the system entropy considerably and therefore, such structures are hardly obtained experimentally. Furthermore, in the highly charged LDHs, the large energy decreases for $x_B < 0.125$ is due to the formation of a stable quasi-two-dimensional hydrogen-bonded network of water (two-dimensional structure without hexagonal, pentagonal, and quadrangular features) and the interlayer contraction. For $x_B > 0.125$, the large increase is due to the subsequent structural transformation to a three-dimensional hydrogen-bonded network along with tilted nitrate ions, accompanied by interlayer expansion that is facilitated at high interlayer filling densities of ions (Fig. 6); this energetically unfavourable transformation precludes nitrate intercalation. Notably, these two- and three-dimensional water structures possibly correspond to SW-I and -II, respectively, as concluded from the TG-DTA analysis. On the other hand, the weakly charged LDHs show structural transformation from a slightly quasi-two-dimensional to a two-dimensional hydrogen-bonded network of water, while maintaining a parallel orientation of the nitrate ions, involving minimal changes in the interlayer distance; this facilitates progressive nitrate intercalation (Fig. 6). This concerted interplay between the interlayer ion configuration, especially the filling density of ions, and structural change of the hydrogen-bonding network of interlayer water governs the nitrate ion storage mechanism and can be precisely tuned by varying the host charge density.

To date, understanding the water structure has been of great importance not only for ion storage layered metal oxides/hydroxides, but also for van der Waals materials, MXenes, and conventional clays. Our results demonstrate that the ion configuration in the interlayer confinement enables precise control of the dynamic structural change in water, considerably improving the ion storage ability of the material. We expect that further tuning of the host

materials will offer a new framework for the optimisation of ion storage-related functionalities.

## Methods

### Synthesis

Reagent-grade $Mg(NO_3)_2 \cdot 6H_2O$, $Al(NO_3)_3 \cdot 9H_2O$, $Na_2CO_3$, NaOH, HCl, and NaCl (Wako Pure Chemical Industries, Ltd.) were used. Highly and weakly charged Mg/Al LDHs with $x = 0.28$ and $0.22$, respectively, were synthesised by the conventional co-precipitation method using an aqueous solution containing $Mg^{2+}$ and $Al^{3+}$ ions as a precursor solution and $NaOH/Na_2CO_3$ aqueous solution as a precipitation agent, followed by hydrothermal treatment at 140 °C for 22 h[24–26]. After drying at 60 °C in air, the samples were subjected to the Cl-exchange treatment method using an acidic chloride aqueous solution to eliminate carbonate contamination (immersed overnight in aqueous HCl (33.0 mM) and NaCl (4.0 M) solutions at room temperature).

### Characterisation

The crystal structures of the obtained LDHs were studied by powder XRD analysis (Miniflex II, Rigaku) performed using a Cu $K\alpha$ radiation ($\lambda = 1.5418$ Å) source operated at 30 kV and 20 mA; the scans were performed for $2\theta$ values ranging from 5° to 80°. The peak positions were calibrated with respect to the external standard (Si powder, Standard Reference Material 640 f, NIST). Ex situ synchrotron powder XRD patterns for analysing the crystal structure of water-soaked LDHs were recorded at beamline BL5S2 at the Aichi Synchrotron Radiation Centre at a beam energy of 12 keV ($\lambda = 0.7$ Å). All samples were dispersed in ultra-pure water and placed in 0.4-mm-diameter Lindemann glass capillaries. The capillaries were rotated during the data collection to obtain uniform diffraction intensities. All samples were measured using the transmission geometry. The X-ray wavelength was calibrated with $CeO_2$ (Standard Reference Material 674b, NIST). The chemical composition of the as-prepared samples was analysed using the ICP-OES (SPS5510, SII NanoTechnology, Hitachi High-Tech Science Corporation, Japan) technique. The TG-DTA (Thermo Plus EVOII TG8120, Rigaku) technique was used to analyse the water content in the as-prepared samples at a heating rate of 10 °C min$^{-1}$. The IR spectra of the as-prepared samples were recorded using a Fourier-transform infrared FT-IR spectrometry system (FT/IR-6100, Jasco) equipped with a smart-endurance single-bounce diamond attenuated total reflection cell (ATR PRO450-S, Jasco). The spectral profiles in the range of 4000–450 cm$^{-1}$ were obtained by adding the results of 100 scans performed at a resolution of 4.0 cm$^{-1}$. Spectral manipulations, such as baseline correction, smoothing, and normalisation, were performed using the Jasco Spectra Manager (version 2) software package. The morphologies and compositional distribution of the as-prepared samples were observed using the FE-SEM (JSM-7400F, JEOL) technique. The SEM instrument was operated at an acceleration voltage in the range of 2−15 kV. Energy-dispersive X-ray spectroscopy (EDX JSM-7000F, JEOL) was also used to analyse the samples.

### Nitrate ion storage performance

To generate the ion-exchange adsorption isotherms for the as-prepared LDHs, the initial concentrations of the nitrate ions were set in the range of 0.2−5.0 mM for the experiments in dilute conditions and to 1.0−200.0 mM for the experiments in concentrated conditions using reagent-grade $NaNO_3$ (Wako Pure Chemical Industries, Ltd.). The powder was mixed with an aqueous solution of nitrate in a capped bottle, and the mixture was subsequently subjected to conditions of automatic shaking (SRR-2, AS ONE, Japan; 150 rpm; overnight). For the dilute condition experiments, the resultant supernatant solution was collected and analysed by ion chromatography (HIS-20A SUPER, Shimadzu, Shimpack IC-SA2 analytical column, Shimpack ICGSA2 guard

column, and a hydrogen carbonate/carbonate eluent) to determine the residual concentration of the nitrate ions. For the concentrated conditions experiments, the powder was collected from the resultant solution and subjected to EDX analysis to determine the adsorbed amount of nitrate ions from the Cl/Al ratio in the LDHs. We confirmed that the pH levels of the testing solution were maintained at ~7.0 before and after the ion-exchange adsorption experiment, ensuring minimal inclusion of carbonate ions. The AEC was defined as the number of moles of adsorbed nitrate per mole of LDH formula unit, $[M^{2+}_{1-x}M^{3+}_x(OH)_2]^{x+}(Cl^-)_x \cdot mH_2O$.

### QCM-D measurements

QCM-D analyses were conducted with a QCM-D instrument (Bioline Scientific) using $Au/SiO_2$ quartz sensors. Prior to the measurements, the $Au/SiO_2$ quartz sensors were cleaned by $UV/O_3$ treatment for 10 min followed by immersion in a 2.0 wt% sodium dodecyl sulfate solution. After 30 min, the sensors were rinsed with ultra-pure water and dried, and then the surface of the sensors was activated to facilitate the fabrication of thin films of LDHs. Subsequently, the LDH thin film was deposited on the sensors by immersing in an aqueous suspension (4.8 mg LDHs per mL) for 1.0 min, followed by drying for 10 min to obtain a mass loading of ~8.5 μg cm$^{-2}$. The high rigidity of the coating was confirmed by the small frequency changes measured in air and water (less than two units of dissipation for all measured overtones). The measurements were performed using 200 mM NaCl and $NaNO_3$ aqueous solution with a flow rate of 300 μL min$^{-1}$ at 25.0 °C. For the investigation of high-viscoelastic materials, the overtone order should be considered for the QCM-D measurement[39]. Therefore, we examined the overtone orders of 1 and 3, and the data were subsequently collected with an overtone order of 3 due to the large noise in the overtone order of 1.

### Computational details

The Vienna ab initio simulation package (VASP) was used with the Perdew-Burke-Ernzerhof formulation of the generalised gradient approximation for the exchange-correlation functional and with the projector-augmented wave method for all calculations[40–43]. An energy cut-off of 500 eV was used (based on the data presented in previous reports)[44–47]. The relaxation of the crystal structure was allowed for the stoichiometric models, and the final energies of the optimised structural geometries were recalculated to correct for the changes in the plane-wave basis during relaxation. For all calculations, a $k$-point mesh was chosen so that the product of the number of $k$-points and the number of atoms in the unit cell was greater than 1000. The convergence criteria were set to $10^{-4}$ eV for electronic self-consistent field calculations and to $10^{-3}$ eV for ionic relaxation. For an appropriate treatment of the long-range van der Waals interactions, the Grimme D2 approach was employed[48,49]. The structural model of LDHs with a $6 \times 6 \times 1$ supercell and 1H stacking sequences was employed in this work, which is analogous to the one-layer hexagonal polytype manasseite, the 1H polytype (Supplementary Fig. 9). In addition, interlayer anions are located beneath the $Al^{3+}$ ions within host layers and water molecules are located in the empty space between interlayer anions. The lattice parameters of the optimised geometries were in good agreement with the experimental observations, with a relative error of <1.5% (Supplementary Fig. 10). For the total energy ($E_{total}$) in Eq. (1), the total energies of $Cl^-$ and $NO_3^-$ were obtained by studying the isolated system with a cubic cell ($a = 20$ Å; for only the gamma point). The energy of the hydrated anions was calculated as the sum of the total energy of ($Cl^-$ and $NO_3^-$) and the hydration enthalpy obtained by experiments in the previous work ($-381$ and $-314$ kJ mol$^{-1}$, respectively)[23]. The total energy of $[Mg_{1-x}Al_x(OH)_2]$ was obtained by setting the interlayer distance to 20 Å to minimise the interlayer interactions.

## Data availability

The experimental data generated in this study are provided in the Supplementary Information/Source Data file. Source data are provided with this paper.

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

## Acknowledgements

This work was supported by the Environment Research and Technology Development Fund 5RF-1902 of the Environmental Restoration and Conservation Agency of Japan, JSPS KAKENHI Grant Numbers 22H00568, 22H04533, 21K14404, and 20H05214, and MEXT Programme for Building Regional Innovation Ecosystem. The SPXRD experiments were conducted at the BL5S2 and BL5S1 beamlines of the Aichi Synchrotron Radiation Centre, Aichi Science & Technology Foundation, Aichi, Japan (Proposal Nos. 2020D6009 and 2020D6008).

## Author contributions

T.S. and K.T. planned the project and supervised all aspects of the research. T.Y. and M.U. performed the synthesis, characterisation, and QCM-D measurement of the compounds. T.S. performed the computational analysis under the supervision of H.S. The framework of this manuscript was constructed by T.S. and H.T. The manuscript was written by T.S. and reviewed by F.H and M.T. with the help of the other authors.

## Competing interests

The authors declare no competing interests.
