## [Peer Review File · Nature Communications]

REVIEWER COMMENTS

Reviewer #1 (Remarks to the Author):

The authors systematically investigated the relationship between the configuration of interlayer ion and water structure in layered double hydroxide, by using in situ QCM-D as well as multiple ex situ experiments and theoretical calculations. The work would shed light on the development of new approaches for improving the nitrate storage performance of LDH or other layered materials. The work is interesting while the following issues are suggested to be figured out to make it more acceptable.

1. It seems the highly charged LDH and weakly charged LDH have the same concentration of H₂O molecules, according to their chemical composition ($[\text{Mg}_{0.72}\text{Al}_{0.28}(\text{OH})_2](\text{Cl})_{0.28}\cdot 0.41\text{H}_2\text{O}$ and $[\text{Mg}_{0.78}\text{Al}_{0.22}(\text{OH})_2](\text{Cl})_{0.22}\cdot 0.41\text{H}_2\text{O}$). Actually, the number of interlayer anions and the interlayer distance would affect the water content. So, possible reasons should be provided here.
2. Why did the authors choose the nitrate ions as the example? Did the authors have considered the hydrated nitrate ions? And the different hydration ability of nitrate and chloride ions?
3. It seems some experimental details have not been provided. For example, how did the authors do the anion exchange experiment? Because the carbonate ions have a very strong combination with the brucite layer of LDH, so, to change the prepared LDH to LDH-Cl and then to the LDH-NO₃, the solved CO₂ should be well removed in all solutions.
4. How many layers did the as-prepared LDH have? And what is the overtone order (n) for the authors collecting the QCM-D? The QCM-D under different overtone order is suggested to be collected to show the change of different depth of LDH, as have been reported by ACS Appl. Mater. Interfaces 2021, 13, 37063–37070.
5. How did the authors exactly determine the filling density of the ions? And how did you differentiate the interlayer ions and surface absorbed ions?

Reviewer #3 (Remarks to the Author):

The manuscript deals with understanding the role of the charge density on the ion-exchange in LDH structures with a special attention to the structuring of interlayer water. The work is carefully performed in terms of both experiments and the DFT modelling. The DFT results fit well to the experimental observation and support well the conclusions. The manuscript is well written and meets standards in the area of ion-exchange materials. The methodology is described in sufficient detail.

The results contribute to understanding of water structure in the clay materials and potentially in the other layered materials. The work brings an important original contribution. The relation of structuring of water with anion storage capacity and charge of the positively-charged hydroxide layers is demonstrated for the first time.

There are a few small points I would like authors address. The dissipation behavior of the LDH layers during QCM-D measurement are correlated by the authors with interlayer interactions. Can the changing interaction between the deposited particles as a result of ion adsorption on their surface lead to a similar effect?

The recently published work (<https://doi.org/10.1021/acsomega.2c01115>) on the DFT/MD modelling of hydration of similar LDH structures and role of hydrogen bonds on the orientation of intercalated anions is not considered in the discussion of modelling results. The computational results obtained there suggest that the hydration state of LDHs has a significant effect on their key properties like interlayer spacing and self-diffusion coefficients of the intercalated anions.

RESPONSE TO REVIEWERS' COMMENTS

Reply to Reviewer #1

We thank reviewer #1 for the positive remarks regarding the novelty of our work.

Comments:

1) It seems the highly charged LDH and weakly charged LDH have the same concentration of H₂O molecules, according to their chemical composition ([Mg_{0.72}Al_{0.28}(OH)₂](Cl)_{0.28}·0.41H₂O and [Mg_{0.78}Al_{0.22}(OH)₂](Cl)_{0.22}·0.41H₂O). Actually, the number of interlayer anions and the interlayer distance would affect the water content. So, possible reasons should be provided here.

Response:

We thank the reviewer for the suggestion. Briefly, the difference in the number of bonds with respect to water molecules (with neighbouring water molecules and anions) in the hydrogen-bonding networks in these two LDHs is a decisive factor in this regard. In detail, water molecules are inserted into the interlayer space, which is not occupied by anions, in order for the laminated structure of the LDHs to be thermodynamically stable. Therefore, weakly charged LDHs is assumed to contain more interlayer water than highly charged LDHs due to a lower interlayer filling density, as indicated in Fig. S12, which is enthalpically advantageous. However, from the results of our theoretical calculations (Fig. 5a) and its interpretation, we proposed that the entropic disadvantage of water structuring along with the incremental interlayer water content determines the final amount of interlayer water. As mentioned above, in weakly charged LDHs, the number of bonds per water molecule required to form a two-dimensional network, i.e., the total number of anions and water molecules, is less than that of highly charged LDHs, which form a three-dimensional network. Therefore, in contrast to the above enthalpic assumption, the increase in interlayer water in the weakly charged LDHs has been suppressed and the amount of interlayer water in two LDHs result in similar values.

Fig. S12 | Illustration of the Connolly surface of the structures presented in Fig. S9 (a) and (b) calculated using the Connolly radius of 1.0 Å, where the water molecules have been removed.

2) Why did the authors choose the nitrate ions as the example? Did the authors have considered the hydrated nitrate ions? And the different hydration ability of nitrate and chloride ions?

Response:

We thank the reviewer for the suggestion. As the reviewer noted, nitrate and chloride ions both have a monovalent negative charge and low nucleophilicity but different molecular structures, resulting in similar interactions between the ions and adsorption site but different hydration behaviour – chloride ions have a large hydration enthalpy [R1] and high ability to structure water, as suggested by the Hofmeister series [R2]. However, the hydration behaviour under confinement of layered materials and how this affects ion storage have been elusive. On this basis, we considered that an ion-exchange reaction system between these two ions is a suitable model system for studying the concerted structuring behaviour of water and ions in the two-dimensional confinement of the layered materials.

[R1] Smith, D. W. Ionic hydration enthalpies. *J. Chem. Educ.* **54**, 540–542 (1977).

[R2] Hofmeister, F. *Arch. Exp. Pathol. Pharmacol.* **24**, 247–260 (1888).

We have added the following sentences ‘On this basis, we considered the separation of nitrate (NO_3^-) ions through the ion-exchange reaction with chloride ions in LDHs’. **in page 4 line 5 (main manuscript)**, as well as ‘More importantly, although both ions have a monovalent negative charge and low nucleophilicity, they possess different molecular structures, resulting in similar interactions between the ions and adsorption site but different hydration behaviours; chloride ions have a large hydration enthalpy [23] and ability to structure water, as suggested by the Hofmeister series [24]. Thus, the ion-exchange reaction system between nitrate and chloride is a suitable model system for studying the concerted structuring behaviour of water and ions in the two-dimensional confinement of the layered materials’. **in page 4 line 10 (main manuscript)**. Along with the above revisions, we further modified the introduction: **page 3 line 11 and page 4 line 15 (main manuscript)**.

In addition, we considered a different hydration enthalpy of the nitrate and chloride ions in the DFT calculations to investigate the favourable direction of the ion-exchange system described by $\text{Cl}^-(\text{s}) + \text{NO}_3^-(\text{l}) \rightleftharpoons \text{Cl}^-(\text{l}) + \text{NO}_3^-(\text{s})$. We applied the experimentally obtained hydration enthalpy [R2] to calculate the binding energy of this system, as described in page 6 line 31 (main manuscript).

3) It seems some experimental details have not been provided. For example, how did the authors do the anion exchange experiment? Because the carbonate ions have a very strong combination with the brucite layer of LDH, so, to change the prepared LDH to LDH-Cl and then to the LDH-NO₃, the solved CO₂ should be well removed in all solutions.

Response:

We thank the reviewer for bringing the carbonate inclusion during the anion exchange experiments to our attention. We conducted experiments by carefully considering the pH condition to eliminate carbonate contamination, as carbonate ions are a major CO₂-relating species in aqueous solutions at pH > 10. In addition, N₂ bubbling during the adsorption test showed little effect on the ion-storage capacity.

According to our experimental procedures, we have added the following sentences ‘The samples were subjected to the Cl-exchange treatment method using an acidic chloride aqueous solution to eliminate carbonate contamination’ **in page 16 line 5 (main manuscript)**, and ‘We confirmed that the pH levels of the testing solution were maintained at approximately 7.0 before and after the ion-exchange adsorption experiment, ensuring minimal inclusion of carbonate ions’. **in page 18 line 8 (main manuscript)**.

4) How many layers did the as-prepared LDH have? And what is the overtone order (n) for the authors collecting the QCM-D? The QCM-D under different overtone order is suggested to be collected to show the change of different depth of LDH, as have been reported by ACS Appl. Mater. Interfaces 2021, 13, 37063–37070.

Response:

We thank the reviewer for the question. The as-prepared LDHs consist of approximately 50 layers, which was calculated by the thickness of the highly charged LDHs (SEM images of LDHs with thickness of approximately 30–40 nm in Fig. R1) and interlayer distance of approximately 0.8 nm determined by XRD patterns. We examined the overtone order of 3 and 1 and the overtone order 1

showed large noise on the profile. Therefore, we decided to adopt 3 in all the QCM-D experiments.

We have added the following sentence ‘For the investigation of high viscoelastic materials, the overtone order should be considered for the QCM-D measurement [40]. Therefore, we examined the overtone orders of 1 and 3, and the data were subsequently collected with an overtone order of 3 due to the large noise in the overtone order of 1’. in page 19 line 5 (main manuscript).

As the reviewer noted, the hierarchical information of ΔD change is important to distinguish the adsorption reaction in the bulk and surface of the LDH particles. In the future, we plan to further investigate the overtone dependence of the QCM-D profile.

Fig. R1 | FE-SEM image of highly charged LDHs for estimating the thickness of the LDH particles.

5) How did the authors exactly determine the filling density of the ions? And how did you differentiate the interlayer ions and surface adsorbed ions?

Response:

We thank the reviewer for the question. We treat the filling density the same as the layer charge density of the host layer due to the charge compensation.

On this basis, we have added the following sentence ‘Therefore, the interlayer filling density of the charge-compensating counter-anions is precisely determined by x ’. in page 4 line 4 (main manuscript).

It is difficult to clearly distinguish between surface adsorbed ions and interlayer ions in the experiment. As described above, the investigation of overtone order dependence of the QCM-D profile is supposed to be effective. However, at present, we believe that the surface contribution is small relative to the interlayer space because the number of adsorption sites in the interlayer space is approximately 50 times larger than the surface, assuming that the number of sites per host layer is the same for the interlayer space and the surface. The remarkable change in the size of the interlayer space along with nitrate adsorption (Fig. 4) supplementarily supports the major contribution of the interlayer adsorption.

Reply to Reviewer #3

We thank reviewer #3 for the positive remarks regarding the novelty of our work.

Comments:

- 1) There are a few small points I would like authors address. The dissipation behavior of the LDH layers during QCM-D measurement are correlated by the authors with interlayer interactions. Can the changing interaction between the deposited particles as a result of ion adsorption on their surface lead to a similar effect?*

Response:

We thank the reviewer for the question. We consider that inter-particle interactions are almost negligible, as the LDH particles are deposited at a sufficient distance from each other on the sensor (Fig. R2a), yet show large viscoelastic changes along with the adsorption reaction. Even when taking into account the influence of some accumulated LDH particles (indicated with yellow dotted circles in Fig. R2a and b), it is difficult to explain the changes in viscoelasticity that increase or decrease nonlinearly (Fig. 3c) since the changes in surface properties, such as surface potential, associated with surface adsorption reactions should vary monotonically.

Fig. R2 | Reposted FE-SEM image of the thin film of highly (a) and weakly charged LDHs (b) deposited on the SiO₂-coated Au electrode (QCM-D sensor), as shown in Fig. S3. The accumulated LDH particles are marked with yellow dotted circles.

2) *The recently published work (<https://doi.org/10.1021/acsomega.2c01115>) on the DFT/MD modelling of hydration of similar LDH structures and role of hydrogen bonds on the orientation of intercalated anions is not considered in the discussion of modelling results. The computational results obtained there suggest that the hydration state of LDHs has a significant effect on their key properties like interlayer spacing and self-diffusion coefficients of the intercalated anions.*

Response:

We thank the reviewer for the suggestion. We re-examined the role of hydrogen bonds on the orientation of intercalated anions in the DFT calculation results.

We have added the following sentences “Second, the interlayer binding energies, the hydrogen-bonding structure associated with orientation of interlayer anions and water molecules, and the interlayer distance [38] for both LDHs with various interlayer anion compositions (x_B) were calculated as a function of the interlayer water content (m).” **in page 11 line 7 (main manuscript);** ‘from parallel to tilted’ **in page 13 line 7 (main manuscript)**, ‘three-dimensional hydrogen-bonded network along with tilted nitrate ions’ **in page 14 line 13 (main manuscript)**, and ‘while maintaining a parallel orientation of the nitrate ions’ **in page 15 line 1 (main manuscript)**.

REVIEWERS' COMMENTS

Reviewer #1 (Remarks to the Author):

The authors have addressed all the concerns the reviewer made with more experimental/theoretical results and explanations. Though the EQCM-D with different overtones have not been provided in the manuscript, it does not affect the main conclusion of the work. So, the reviewer recommend the acceptance of the work.

Reviewer #3 (Remarks to the Author):

The authors have properly addressed the critical points in the revised version. Therefore, I recommend the manuscript for publication.

RESPONSE TO REVIEWERS' COMMENTS

Reviewer #1 (Remarks to the Author):

Comments:

The authors have addressed all the concerns the reviewer made with more experimental/theoretical results and explanations. Though the EQCM-D with different overtones have not been provided in the manuscript, it does not affect the main conclusion of the work. So, the reviewer recommend the acceptance of the work.

Response:

We thank the reviewer for recommending the acceptance of our manuscript. We appreciate the reviewer for giving their precious time to provide suggestions. The reviewer's comments significantly improved our manuscript.

Reviewer #3 (Remarks to the Author):

Comments:

The authors have properly addressed the critical points in the revised version. Therefore, I recommend the manuscript for publication.

Response:

We thank the reviewer for recommending the acceptance of our manuscript. We appreciate the reviewer for giving their precious time to provide suggestions. The reviewer's comments significantly improved our manuscript.